# Peanut Shell Extract Improves Mitochondrial Function in db/db Mice via Suppression of Oxidative Stress and Inflammation

**DOI:** 10.3390/nu16131977

**Published:** 2024-06-21

**Authors:** Hemalata Deshmukh, Julianna M. Santos, Matthew Bender, Jannette M. Dufour, Jacob Lovett, Chwan-Li Shen

**Affiliations:** 1Department of Pathology, Texas Tech University Health Sciences Center, Lubbock, TX 79430, USA; hemdeshm@ttuhsc.edu (H.D.); julianna.santos@ttuhsc.edu (J.M.S.); jacob.lovett@utexas.edu (J.L.); 2Department of Medical Education, Texas Tech University Health Sciences Center, Lubbock, TX 79430, USA; matthew.bender@ttuhsc.edu (M.B.); jannette.dufour@ttuhsc.edu (J.M.D.); 3Department of Cell Biology and Biochemistry, Texas Tech University Health Sciences Center, Lubbock, TX 79430, USA; 4Center of Excellence for Integrative Health, Texas Tech University Health Sciences Center, Lubbock, TX 79430, USA; 5Obesity Research Institute, Texas Tech University, Lubbock, TX 79401, USA; 6Center of Excellence for Translational Neuroscience and Therapeutics, Texas Tech University Health Sciences Center, Lubbock, TX 79430, USA

**Keywords:** bioactive compounds, luteolin, mitochondria, tissues, diabetes, mice

## Abstract

Accumulating evidence shows a strong correlation between type 2 diabetes mellitus, mitochondrial dysfunction, and oxidative stress. We evaluated the effects of dietary peanut shell extract (PSE) supplementation on mitochondrial function and antioxidative stress/inflammation markers in diabetic mice. Fourteen db/db mice were randomly assigned to a diabetic group (DM in AIN-93G diet) and a PSE group (1% wt/wt PSE in AIN-93G diet) for 5 weeks. Six C57BL/6J mice were fed with an AIN-93G diet for 5 weeks (control group). Gene and protein expression in the liver, brain, and white adipose tissue (WAT) were determined using qRT-PCR and Immunoblot, respectively. Compared to the control group, the DM group had (i) increased gene and protein expression levels of DRP1 (fission), PINK1 (mitophagy), and TNFα (inflammation) and (ii) decreased gene and protein expression levels of MFN1, MFN2, OPA1 (fusion), TFAM, PGC-1α (biogenesis), NRF2 (antioxidative stress) and IBA1 (microglial activation) in the liver, brain, and WAT of db/db mice. Supplementation of PSE into the diet restored the DM-induced changes in the gene and protein expression of DRP1, PINK1, TNFα, MFN1, MFN2, OPA1, TFAM, PGC-1α, NRF2, and IBA1 in the liver, brain, and WAT of db/db mice. This study demonstrates that PSE supplementation improved mitochondrial function in the brain, liver, and WAT of db/db mice, in part due to suppression of oxidative stress and inflammation.

## 1. Introduction

Type 2 diabetes mellitus (T2DM) is rapidly becoming the most prevalent metabolic disorder in developed nations [1,2]. Emerging evidence shows a strong association between T2DM, mitochondrial dysfunction, and oxidative stress [3,4]. Mitochondria, a key player in energy metabolism, is one of the primary targets of T2DM at the intracellular level [5,6]. Glucose is a source of energy in most tissues [7], with liver and adipose tissue serving as modulators of energy balance and glucose homeostasis [8]. In T2DM, glucose uptake disorder can lead to disruption of cellular energy metabolism and thus hamper mitochondrial function [5]. The common pathological change is increased blood glucose levels or hyperglycemia, leading to serious damage to tissues, such as the liver, brain, and adipose tissue [5,9]. Even now, the molecular mechanism underlying damage to the mitochondrial function in T2DM is still unclear [5]. Therefore, more research is warranted to study mitochondrial dysfunction in T2DM, specifically in different tissues.

Mitochondrial dynamics/interactions are undergoing a coordinated process of fission and fusion [10]. Mitochondrial fission is regulated by dynamin-related protein 1 (DRP1) and mitochondrial fission 1 (FIS1), whereas mitochondrial fusion is facilitated by a range of proteins, such as mitofusin-1 (MFN1) and mitofusin-2 (MFN2). Additionally, optic atrophy 1 (OPA1) serves as a dynamin-related GTPase involved in the control of mitochondrial dynamics. PTEN-induced putative kinase 1 (PINK1) is involved in the process of mitophagy [4]. Excessive oxidative stress within mitochondria plays a pivotal role in mitochondrial dysfunction across various organ systems, including the liver, brain, and adipose tissue [11]. Mitochondrial dysfunction and increased reactive oxygen species (ROS) production are associated with inflammation activation and insulin resistance development during the progression of type 2 diabetes mellitus (T2DM) [12,13].

Peanut shell (by-product of the peanut) is rich in phenolic compounds, flavonoids, and lignans, which possess antioxidant capacities to scavenge ROS and suppress oxidation [14]. Intriguingly, peanut shell extract (PSE) was shown to suppress alpha-amylase activity, an enzyme that digests carbohydrates and starch into smaller poly-/mono-saccharides during digestion, indicating PSE’s anti-T2DM potential [15]. PSE has been shown to have hypoglycemic effects in high-fat diet/streptozotocin (HFD/STZ)-induced diabetic rats [16]. In addition, luteolin, the major bioactive compound in PSE, has been reported to lessen insulin resistance by activating AMPK signaling in KK-A(y) mice, a model of spontaneously hyperglycemic, hypercholesterolemic, and hypertriglyceridemic animals [17,18]. However, no study has ever investigated the effects of PSE in the db/db diabetic mice with the aspects of oxidative stress and mitochondrial dysfunction. Thus, this study evaluated the effects of PSE supplementation on mitochondrial function markers along with antioxidative stress/inflammation markers in three major tissues, namely the liver, brain, and white adipose tissue (WAT) in db/db mice. In this study, a Cg-*Dock7^m^* +/+ *Lepr^db^*/J (db/db) model was used to model phases I to III of T2DM and obesity (https://www.jax.org/strain/000642) (accessed on 1 October 2023). We hypothesized that supplementation of PSE in the diet would mitigate T2DM-induced mitochondrial dysfunction in the liver, brain, and WAT. Such beneficial effects of PSE in diabetic mice would be, in part, mediated via suppression of mitochondrial oxidative stress and inflammation.

## 2. Materials and Methods

### 2.1. Animals and Treatments

Fourteen 5-week-old male homozygous BKS.Cg-*Dock7^m^* +/+ *Lepr^db^*/J (db/db) mice (strain #: 000642) were purchased from the Jackson Laboratory, Bar Harbor, ME, USA. An additional 6 males C57BL/6J (5-week-old) were used for the control group (Jackson Laboratory, strain #: 000664).

The animals were housed (2–3 mice per cage) at a constant temperature of (22 ± 2) °C and humidity (of 55 ± 5) % with a 12 h light/dark cycle. Mice were acclimatized to the AIN-93G diet (catalog number: D10012G, Research Diet, Inc., New Brunswick, NJ, USA). After 1-week acclimation, the db/db mice were randomly assigned to the diabetic control group (DM) or PSE group (PSE at 1.0% wt/wt in AIN-93G diet). The control C57BL/6J mice were fed with an AIN-93G diet throughout the study. Food and water were provided ad libitum throughout the 5-week study feeding period. The PSE contained 20% luteolin concentration (Sabinsa Corporation, East Windsor, NJ, USA). All procedures were approved by the Institutional Animal Care and Use Committee (IACUC, protocol #22017). All the experiments were conducted following the relevant guidelines and regulations. Throughout the study period, there were no unexpected adverse events observed in all animals.

### 2.2. Sample Collection

Prior to sample collection, the animals were fasted for 4 h. Animals were anesthetized with isoflurane and euthanized. Liver, brain, and WAT were harvested, immersed in liquid nitrogen, and kept at −80 °C before analysis. In order to minimize potential confounders, such as the order of treatment and measurements, we performed all lab analyses on all animals in the same order and on the same day.

### 2.3. RNA Extraction and qRT-PCR

Total RNA was isolated from the liver, brain, and WAT using the RNAzol RT (RN190, Molecular Research Center Inc., Cincinnati, OH, USA) and a BAN ratio 1:200 (BN191, Molecular Research Center, Cincinnati, OH, USA). RNA isolation and qRT-PCR for respective gene expression (Appendix A) were described in our previous work [19]. Gene expressions were normalized using β-actin as a housekeeping gene. The expression of genes was calculated using the following formula: 2^−(ΔCT × 1000)^ [20].

### 2.4. Protein Extraction and Western Blot Analysis

Proteins were extracted from the liver, brain, and WAT using RIPA lysis buffer with protease and phosphatase inhibitors (Cat # 78440, Thermo Scientific, Waltham, MA, USA). The tissues were homogenized on ice using a sonicator, and the protein supernatant was collected after centrifugation. Protein concentration was determined using a BCA protein assay kit (Cat #23227, Rockford, IL, USA). For immunoblot analysis, 40 µg of proteins were denatured and separated by SDS-PAGE gel (Mini-PROTEAN^®^ TGX™ Precast Protein Gels Cat #4561093 Bio-Rad Laboratories, Hercules, CA, USA). Proteins were then transferred (Cat #1704150 Bio-Rad Laboratories, Hercules, CA, USA) to a PVDF membrane (Cat #1704273, Bio-Rad Laboratories, Hercules, CA, USA) and blocked with 5% BSA. Specific antibodies were used to probe the separated proteins overnight at 4 °C or for 2 h at room temperature. After washing, protein complexes were detected using HRP-conjugated antibodies and chemiluminescence reagents (Pierce™ ECL Immunoblotting Substrate, Cat #32106, Rockford, IL, USA). Band exposures were kept within the linear range, and results were normalized to β-actin values. Densitometric analysis was performed using ImageJ software (ImageJ-win64). Antibodies and their respective dilutions are listed in Appendix A.

### 2.5. Statistical Analysis

The data are presented as the mean ± standard error of the mean (SEM) and analyzed using one-way ANOVA followed by Uncorrected Fisher’s LSD test with GraphPad Prism 9 (GraphPad Software, San Diego, CA, USA). Significance levels are indicated as follows: * for *p* < 0.05, ** for *p* < 0.005, *** for *p* < 0.0005, **** for *p* < 0.00005 and # for 0.05 < *p* < 0.1.

## 3. Results

### 3.1. Mitochondria Fission and Fusion Markers

Effects of PSE on mitochondrial fission-associated genes and proteins [DRP1 (Figure 1) and FIS1 (Figure 2) were measured in the liver, brain, and WAT of mice. Compared to the control animals, the DM animals exhibited elevated gene and protein expression levels of DRP1 in all three tissues. PSE supplementation significantly reduced DM-induced DRP1 gene and protein expression (Figure 1). In contrast to the DRP1 observation, the results of FIS1 (a mitochondrial adaptor of DRP1) showed that (1) the DM mice had lower gene and protein expression levels of FIS1 in all three tissues than those in the control mice, and (ii) the PSE mice had higher gene and protein expression of FIS1 in the studied tissues than those in the DM mice (Figure 2).

Figure 3, Figure 4 and Figure 5 show the effects of PSE on mitochondrial fusion genes and protein expression of MFN1, MFN2, and OPA1, respectively, in the liver, brain, and WAT of mice. In comparison to the control group, the DM group showed decreased gene expression of MFN1 (Figure 3) and MFN2 (Figure 4) in the liver, brain, and WAT, respectively. PSE administration significantly increased both MFN1 and MFN2 gene expression in the db/db mice. The protein expression levels of MFN1 and MFN2 were consistent with the respective gene expression results accordingly, except for MFN2 protein expression in the liver showing no difference between the control group and the DM group.

Figure 5 illustrates the impact of PSE on gene and protein expression of OPA1 (a dynamin-related GTPase that controls mitochondrial fusion) in the collected tissues of mice. Supplementation of PSE to the diet significantly increased DM-suppressed OPA1 expression in the liver, brain, and WAT of db/db mice.

### 3.2. Mitochondria Biogenesis Markers

We also evaluated the effects of PSE on PGC-1α and TFAM, which are associated with mitochondrial biogenesis in the liver, brain, and WAT of mice. Compared to the control group, the DM group showed decreased gene/protein expression levels of PGC-1α (Figure 6) and TFAM (Figure 7) in all tissues, except for PGC-1α in the liver with no difference in gene expression (*p* > 0.05) and protein expression (0.05 < *p* < 0.1). PSE supplementation significantly increased both gene and protein expression levels of PGC-1α (Figure 6) and TFAM (Figure 7) in these tissues.

### 3.3. Mitophagy Marker

Figure 8 presents the effects of PSE on PINK1 (a mitochondrial kinase gene acting as a mitophagy marker) in the liver, brain, and WAT of mice. In comparison to the control group, the DM group exhibited induced PINK1 gene/protein expression levels in the liver, brain, and WAT. Administration of PSE suppressed such DM-induced PINK1 gene and protein expression in the above tissues of db/db mice.

### 3.4. Antioxidative Stress and Inflammation Markers

We evaluated the impacts of PSE supplementation on the markers of antioxidative stress (NRF2)/pro-inflammation (TNFα) in the liver, brain, and WAT of mice. Relative to the control mice, the DM mice had decreased NRF2 gene (liver, brain, and WAT) and protein expression levels (brain and WAT) in the db/db mice (Figure 9). PSE supplementation into the AIN-93G diet significantly increased DM-suppressed NRF2 gene and protein expression levels in db/db mice (Figure 9). As anticipated, the DM group exhibited elevated TNFα gene expression across all three tissues, accompanied by increased TNFα protein expression in the brain and WAT of mice (Figure 10). Administration of PSE to db/db mice significantly suppressed the TNFα gene and protein expression in the studied tissues (Figure 10).

### 3.5. Microglia Marker

Figure 11 shows that (i) compared to the control mice, the DM mice had lower IBA1 gene expression levels in the liver, brain, and WAT of mice; (ii) PSE supplementation significantly increased IBA1 gene and protein expression in db/db mice; and (iii) the patterns of protein expression were consistent with those at gene level in studied tissues of mice, except for IBA1 protein expression in liver with no difference between the control mice and DM mice (0.05 < *p* < 0.1) (Figure 11).

## 4. Discussion

We successfully employed this Cg-*Dock7^m^* +/+ *Lepr^db^*/J (db/db) mouse model to study the impacts of PSE on the mitochondrial function in three major tissues (brain, liver, and WAT) in db/db mice. Published studies have shown a link between mitochondrial dysfunction and T2DM [21,22]. The brain maintains whole-body energy homeostasis by adjusting energy availability as well as energy input. Mitochondria play a vital role in supplying energy to the brain. For example, mitochondria respond to the availability of nutrients via fusion or fission in order to maintain energy homeostasis; however, metabolic diseases like T2DM can disrupt these processes [23]. Brain mitochondrial function is maintained, in part, due to a subtle balance between mitochondrial fusion-fission, biogenesis, and mitophagy during the early stages of T2DM [24].

In this study, relative to the control mice, the DM mice exhibited compromised mitochondrial dynamics, as demonstrated by increased fission (DRP1) and mitophagy (PINK1) as well as decreased fusion (MFN1, MFN2, and OPA1) and biogenesis (PGC-1α and TFAM) gene and protein expression levels in the brain of db/db mice. Our findings corroborate previous studies [25,26]. Edwards et al. reported that the BKS.Cg-m^+/+^ Lepr^db^/J, BKS-db/db mice had hyperglycemia that caused an excess of mitochondrial fission (DRP1) and damaged mitochondria in the neurons of diabetic mice with neuropathy [25]. Maneechote et al. described that the HFD-induced pre-T2DM rats experienced raised plasma insulin and homeostatic model assessment–insulin resistance (HOMA-IR) index as well as impaired function of mitochondria in the brain of animals [26]. Our study is the first study to demonstrate the potent effects of PSE on improving mitochondrial dynamics [fission (DRP1) and fusion (MFN1, MFN2, and OPA1)] in the brains of db/db mice. Interestingly, mitochondrial dysfunction mediated by DRP1 is associated with synaptic injury in the brain, especially hippocampal neurons, of db/db animals, suggesting that the identification of DRP1-mediated mitochondrial dysfunction suggests a potential therapeutic target for addressing diabetic brain complications [27]. Future studies are warranted to investigate how PSE affects mitochondrial dynamics in the neurons of different regions of the brain, such as hippocampal neurons.

Similar to the brain findings in db/db mice, dysfunctional mitochondria were noted in the liver and WAT of diabetic mice. Such changes in gene/protein expression levels of fission mediators (DRP1) and fusion mediators (MFN1, MFN2, and OPA1) observed in both tissues of diabetic mice align with findings from prior studies on hyperglycemic-induced mitochondrial fission [19] and fragmentation [28,29]. The present investigation is the pioneering study showcasing the efficacy of PSE to restore the DM-induced changes in both fission and fusion gene/protein expression in diabetic mice.

The dynamics of mitochondrial fission and fusion are critical for cellular function and health. The balance between these processes is significantly influenced by the levels and activity of proteins such as DRP1 and FIS1. In the current study, the elevated FIS1 expression may aid in stress-induced mitochondrial quality control or facilitate the removal of dysfunctional mitochondria [30]. In addition, we observed the opposite variation of DRP1 and FIS1 in the liver, brain, and WAT, suggesting more efficient mitochondria and/or less mitochondrial fission activity as a result of less mitochondrial fragmentation [31].

The development of T2DM is correlated with changes in mitochondrial biogenesis [32]. A number of genes and proteins, such as PGC-1α and TFAM, are closely associated with T2DM development [32]. PGC-1α functions as a transcriptional coactivator, serving as a central regulator of mitochondrial biogenesis and function, which includes oxidative phosphorylation and ROS detoxification [33]. PGC-1α is highly expressed in major glucose metabolic tissues, such as the brain, liver, and WAT, with high energy demands [34,35]. PGC-1α plays a pivotal role in glucose homeostasis and the regulation of T2DM across diverse organs, including the liver, brain, WAT, muscle, pancreas, and kidney [34,35]. In the present study, the results revealed that db/db mice exhibited decreased PGC-1α and NRF2, as well as increased TNF-α gene and protein expression, which is supported by Rius-Pérez’s report that under inflammatory conditions, decreased levels of PGC-1α lead to the downregulation of mitochondrial antioxidant gene expression, resulting in oxidative stress and promoting the activation of nuclear factor kappa B in the context of T2DM [33]. Notably, our results that PSE supplementation increased PGC-1α gene/protein expression in the liver and WAT of diabetic mice agree with Fang’s work using dietary bioactive polyphenols (Celastrol) in the liver and WAT of obese mice with hyperglycemia and insulin resistance [36].

Besides PGC-1α, TFAM also plays an important role in mitochondrial biogenesis in T2DM by controlling packaging, stability, and replication [37]. In this study, we showed that the diabetic mice had decreased TFAM gene and protein expression levels in the studied tissues. Such observation is comparable with Santos’ work showing TFAM’s levels are lower in the retinal mitochondria in animals with diabetes [24]. Furthermore, this study is the first study to report that PSE benefits T2DM via improving mitochondrial biogenesis, as shown by elevated expression levels of PGC-1α and TFAM gene/protein in three major tissues of db/db mice. The above findings are supported by previous studies using bioactive compounds in mouse hepatocytes [38] and human kidney cells (HK-2) [39]. Chowanadisai et al. reported that pyrroloquinoline quinone (a bioactive polyphenol antioxidant) stimulates mitochondrial biogenesis of mouse hepatocytes [38]. Ho et al. showed that oyster-derived extract shields renal tubular HK-2 cells from oxidative stress by bolstering mitochondrial biogenesis, as evidenced by the induced mRNA expression levels of PGC-1α and TFAM [39].

Mitophagy controls mitochondrial quality in T2DM pathology [40]. Both lowered and excessive mitophagy are involved in the pathogenesis of T2DM [40]. The PINK1-associated autophagic process is a molecular mechanism for mitophagy that may result in mitochondrial dysfunction and oxidative/nitrative stress [41,42]. A PINK1-mediated mitophagy pathway may illustrate the development or management of T2DM. Under T2DM conditions, mitochondria undergo depolarization and disruption of regular proteolytic processing of PINK1, consequently triggering the initiation of mitophagy [4]. In this study, the db/db mice had significantly increased gene and protein expression of PINK1 in the liver, brain, and WAT, indicating increased mitophagy through a PINK1-mediated pathway in diabetic mice. Our discovery of increased mitophagy in the above-mentioned tissues of diabetic mice aligns with previous research conducted on diabetic animal models [19,43,44,45]. Intriguingly, the findings that PINK1 levels were significantly suppressed in all three measured tissues by PSE administration suggest that mitophagy decreased in PSE-treated diabetic mice.

Mitochondria serve as the primary site for generating energy in the form of ATP and are also the primary source of ROS production. Alterations in the balance between fusion and fission processes can lead to elevated oxidative stress and inflammation in mitochondria and various tissues in T2DM progression, contributing to insulin resistance [4,28]. In the current study, the observations of (i) increased fission and fragmentation of mitochondria in the brain, liver, and WAT, (ii) decreased NRF2 (an antioxidative stress marker), and (iii) increased TNF-α (an inflammation marker) corroborate Kabra’s study that increased fission and fragmentation of mitochondria have been associated with hyperglycemia-induced ROS overproduction in both mouse and human islets [28]. Thus, to ameliorate mitochondrial dysfunction observed in T2DM, it’s crucial to effectively manage/mitigate oxidative stress and inflammation to ensure proper regulation of mitochondrial quality.

In this study, the findings that PSE supplementation increased NRF2 expression in the liver, brain, and WAT of db/db mice are supported by the previous studies showing the beneficial role of NRF2 induction in cancerous [46] and non-cancerous diseases [47] via the NRF2/KEAP1 (Kelch Like ECH Associated Protein 1) pathway [48,49,50]. Under oxidant stimuli, activated NRF2 detaches from KEAP1 (a sensor for oxidative and electrophilic stresses) and migrates into the nucleus to bind the antioxidant response element (ARE) sequences in the promoter region of antioxidant enzymes (i.e., heme oxygenase, catalase, glutathione peroxidase, and superoxide dismutase). Thus, NRF2 activation has been suggested as a strategy to ameliorate mitochondrial dysfunction by bioactive compounds in T2DM [48,50].

It is worth noting that NRF2 modulates WAT to maintain glucose and lipid homeostasis, including the process of adipogenesis [51]. In this study, PSE supplementation reversed the gene and protein expression of NRF2 and TNF-α in three studied tissues (brain, liver, and WAT) of db/db mice. This study supports PSE’s antioxidative/anti-inflammatory effects, agreeing with previous studies on PSE and luteolin (the most abundant and potent bioactive compounds in PSE) [52,53]. Moreover, we found that treatment with PSE improved HOMA-IR, indicating it also decreased insulin resistance. Furthermore, in the present study, we noted that there is a direct relation between NRF2 and PGC-1α in mitochondrial function and homeostasis of diabetic mice by PSE. Our observation corroborates with Deng’s study that the NRF2/PGC-1α pathway is involved in the regulation of mitochondrial function and homeostasis in ovarian cancer cells [54].

Although a strict classification of microglial roles remains elusive in T2DM, emerging evidence suggests diabetes influences microglial phenotypic states via modulating IBA1 (a sensitive marker for microglia/macrophage) activation in the progression of T2DM [55,56]. In this study, we showed PSE supplementation induced T2DM suppressed IBA-1 gene and protein expression in the studied tissues of diabetic mice. With that being said, future studies are warranted to illustrate how PSE and its bioactive compounds affect microglial activation/function in the progression of T2DM. IBA1 is traditionally known as a marker for microglia, the resident macrophages of the central nervous system [57]. However, IBA1 is also expressed in other types of macrophages and cells involved in the immune response, not just microglia. Understanding its role in non-neuronal tissues, particularly in the context of liver and adipose tissue, provides insights into its broader biological functions and its involvement in various pathological conditions [58,59].

In this study, we learned that the brain and WAT exhibit more susceptibility to changes compared to the liver due to their central roles in metabolism, endocrine regulation, and immune response [60,61,62]. Metabolically, the brain’s high energy demands for cognitive functions and WAT’s involvement in energy storage render them sensitive to alterations in nutrient availability and hormonal signaling. Additionally, their complex regulation by neural and endocrine pathways, coupled with their immunological activities, makes them responsive to environmental and physiological stimuli. Furthermore, the extensive vascularization of these tissues increases their exposure to systemic factors, exacerbating their vulnerability to metabolic disturbances and inflammatory insults. In contrast, the liver’s robust regenerative capacity and metabolic adaptability afford it greater flexibility to perturbations. Overall, the complex interplay of metabolic, endocrine, and immunological factors underscores the increased susceptibility of the brain and WAT to changes compared to the liver [60,61,62].

We noted that this study focuses on the overall function of mitochondria rather than solely on mitophagy. In future studies, in addition to the overall function of mitochondria and PINK1 level, investigating other mitochondrial markers, such as TOMM20, TIMM23, LC3, and LAMP1/2, would further elucidate the role of PSE on mitophagy [63]. Furthermore, the published studies reported that luteolin (a bioactive compound in PSE) has protected cells from oxidative stress-induced cell death and suppressed inflammation in a variety of oxidative stress-cell culture models [64,65,66]. Future study is also warranted to confirm the activity of NRF2 in diabetic mice by assessing other antioxidant enzymes (i.e., HO-1 and catalase) in tissue samples to determine whether the NRF2 increase was able to mitigate ROS production in diabetic mice due to PSE [67,68,69].

## 5. Conclusions

PSE supplementation into the diet improved mitochondrial function in the brain, liver, and WAT of diabetic mice. Such improvement in mitochondrial function may be, in part, due to the inhibition of oxidative stress/inflammation in diabetic mice. The PSE used in this study is a combination of all PSE bioactive compounds such as luteolin. In future studies, further extract of the PSE should be evaluated to clarify the main anti-diabetic components in T2DM management.

## Figures and Tables

**Figure 1 nutrients-16-01977-f001:**
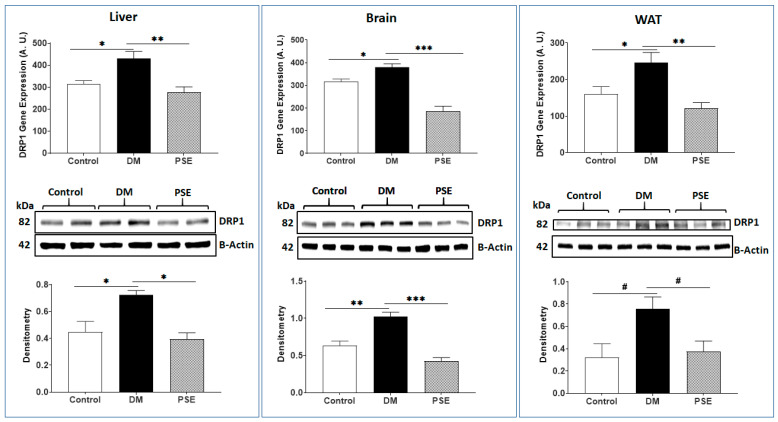
The impact of PSE on DRP1 gene and protein expression in the liver, brain, and WAT of mice. The data are expressed as mean ± SEM. There were 6–7 samples per group (*n* = 6–7). Statistical analysis was conducted using one-way ANOVA followed by Uncorrected Fisher’s LSD test using GraphPad Prism 9. Data are presented as follows: * *p* < 0.05, ** *p* < 0.005, *** *p* < 0.0005 and # 0.05 < *p* < 0.1.

**Figure 2 nutrients-16-01977-f002:**
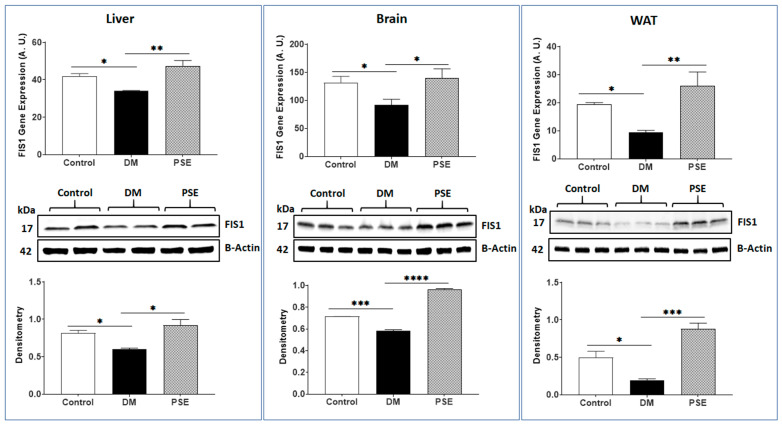
The influence of PSE on FIS1 gene/protein expression in the liver, brain, and WAT of mice. The data are expressed as mean ± SEM. *n* = 6–7 per group. Statistical analysis was conducted with one-way ANOVA and Uncorrected Fisher’s LSD test using GraphPad Prism 9. The data are presented as follows: * *p* < 0.05, ** *p* < 0.005, *** *p* < 0.0005, and **** *p* < 0.00005.

**Figure 3 nutrients-16-01977-f003:**
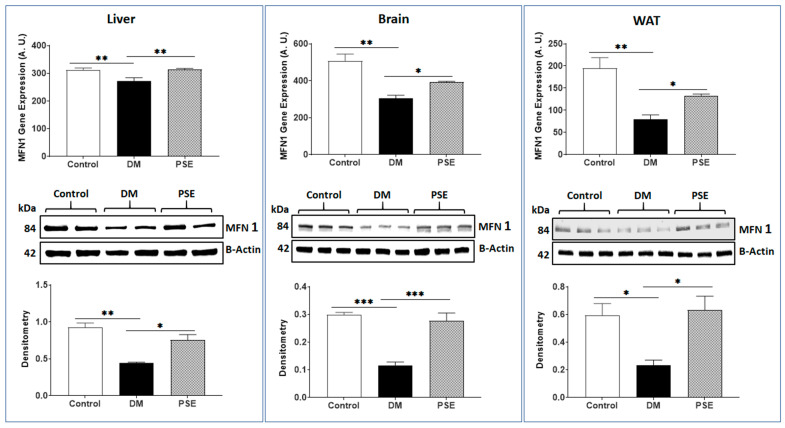
Result of PSE administration in mice on f MFN1 gene as well as protein expression in the liver, brain, and WAT tissue. Data are stated as mean ± SEM. *n* = 6–7 per group. Statistics were carried out by one-way ANOVA followed by Uncorrected Fisher’s LSD using GraphPad Prism 9. The data are shown as follows: * *p* < 0.05, ** *p* < 0.005, and *** *p* < 0.0005.

**Figure 4 nutrients-16-01977-f004:**
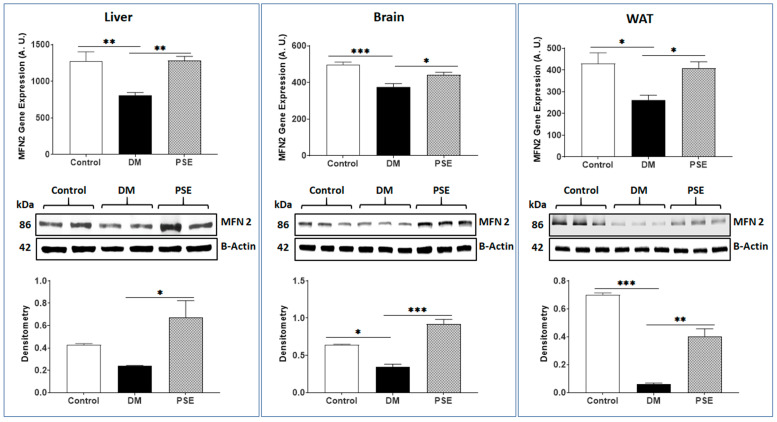
PSE supplementation modulates expression of MFN2 at gene and protein levels in different tissues of mice. Data are expressed as mean ± SEM. *n* = 6–7 per group. Data analysis was carried out by using one-way ANOVA followed by Uncorrected Fisher’s LSD with GraphPad Prism 9. Data is stated as * *p* < 0.05, ** *p* < 0.005, and *** *p* < 0.0005.

**Figure 5 nutrients-16-01977-f005:**
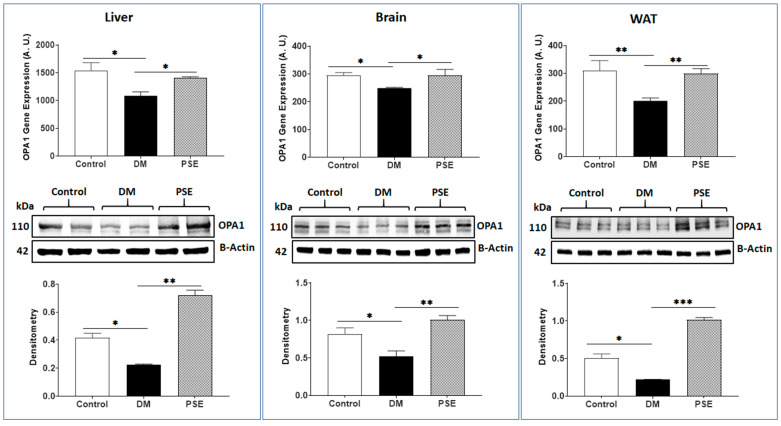
PSE administration showed an effect on the expression of OPA1 in the liver, brain, and WAT tissue of mice. The data are presented as mean ± SEM with 6–7 samples per group (*n* = 6–7). Statistical analysis was performed using one-way ANOVA followed by Uncorrected Fisher’s LSD test using GraphPad Prism 9. Significance levels are denoted as follows: * for *p* < 0.05, ** for *p* < 0.005, and *** for *p* < 0.0005.

**Figure 6 nutrients-16-01977-f006:**
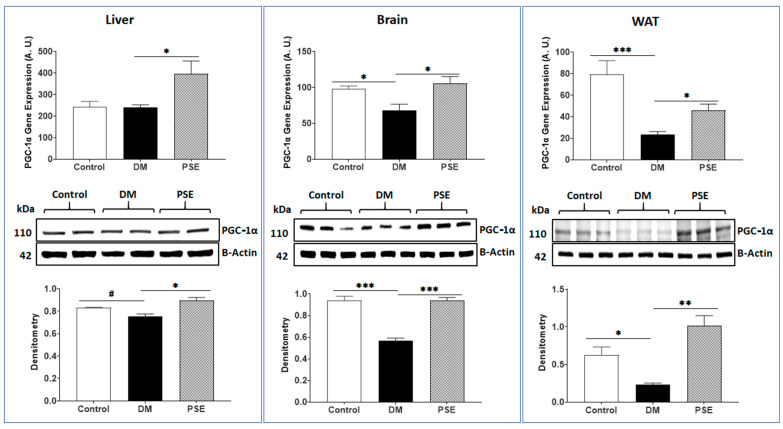
The impact of PSE on PGC-1α gene and protein expression in the liver, brain, and (WAT) of mice. Data are expressed as mean ± SEM with 6–7 samples per group (*n* = 6–7). Statistical analysis was conducted using one-way ANOVA followed by Uncorrected Fisher’s LSD test using GraphPad Prism 9. Significance levels are indicated as follows: * for *p* < 0.05, ** for *p* < 0.005, *** for *p* < 0.0005, and # for 0.05 < *p* < 0.1.

**Figure 7 nutrients-16-01977-f007:**
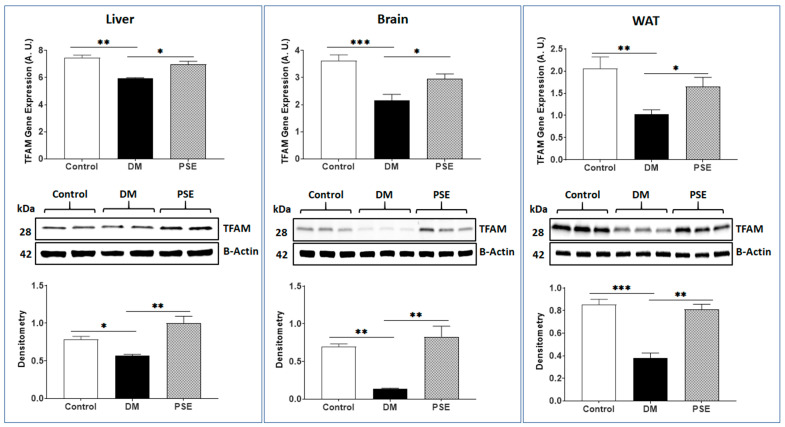
The impact of PSE on expression of TFAM at gene/protein level in different tissues of mice. Data are presented as mean ± SEM with 6–7 samples per group (*n* = 6–7). Statistical analysis was performed using one-way ANOVA and Uncorrected Fisher’s LSD test using GraphPad Prism 9. Significance levels are denoted as follows: * for *p* < 0.05, ** for *p* < 0.005, and *** for *p* < 0.0005.

**Figure 8 nutrients-16-01977-f008:**
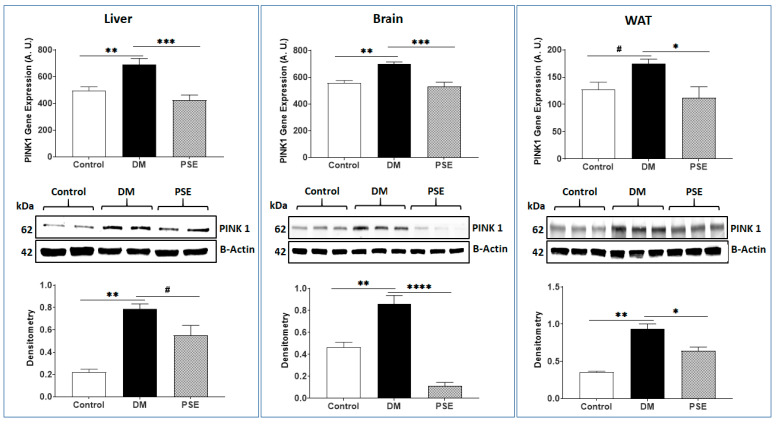
Effect of PSE on PINK1 gene and protein expression in the liver, brain, and WAT of mice. The data are presented as mean ± SEM, *n* = 6–7 per group. Statistical analysis was carried out by using one-way ANOVA followed by Uncorrected Fisher’s LSD with GraphPad Prism 9. The analysis is presented as * *p* < 0.05, ** *p* < 0.005, *** *p* < 0.0005, **** *p* < 0.00005 and # 0.05 < *p* < 0.1.

**Figure 9 nutrients-16-01977-f009:**
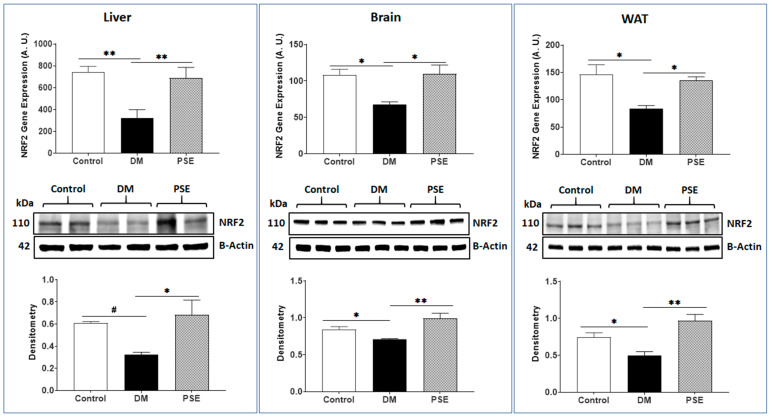
PSE administration in mice changes the NRF2 gene and protein expression in the liver, brain, and WAT tissues. The data are expressed as mean ± SEM. *n* = 6–7 per group. Statistical significance was analyzed by one-way ANOVA followed by Uncorrected Fisher’s LSD using GraphPad Prism 9. The data is expressed as * *p* < 0.05, ** *p* < 0.005, and # 0.05 < *p* < 0.1.

**Figure 10 nutrients-16-01977-f010:**
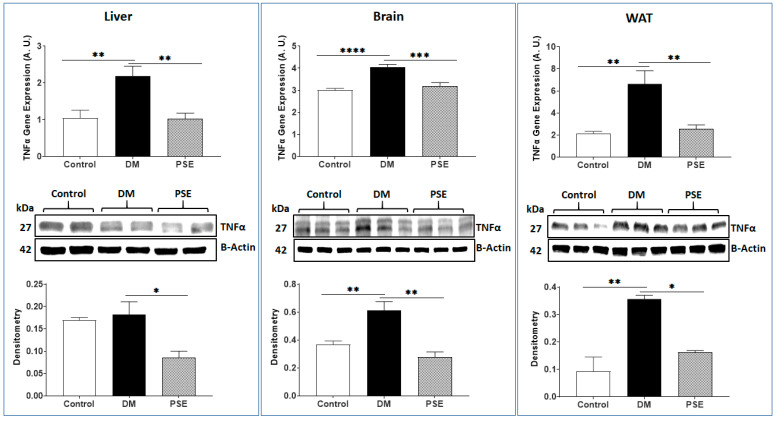
The effect of PSE on TNFα gene and protein expression in different tissues of mice. The data are expressed as mean ± SEM. *n* = 6–7 per group. Statistical analysis was carried out by one-way ANOVA followed by Uncorrected Fisher’s LSD with GraphPad Prism 9. * *p* < 0.05, ** *p* < 0.005, *** *p* < 0.0005, and **** *p* < 0.00005.

**Figure 11 nutrients-16-01977-f011:**
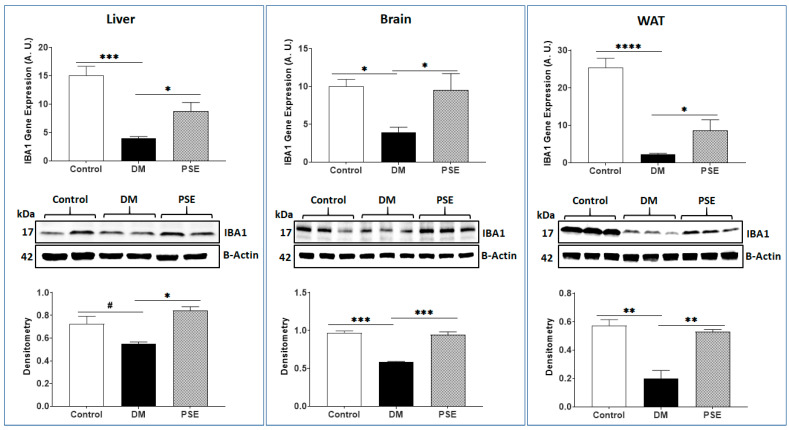
PSE administration in mice leads to alteration in gene and protein expression of IBA1 in various tissues. The data are expressed as mean ± SEM. *n* = 6–7 per group. The data were analyzed by one-way ANOVA followed by Uncorrected Fisher’s LSD using GraphPad Prism 9. * *p* < 0.05, ** *p* < 0.005, *** *p* < 0.0005, **** *p* < 0.00005, and # 0.05 < *p* < 0.1.

## Data Availability

The original contributions presented in the study are included in the article/Appendix A, further inquiries can be directed to the corresponding author.

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
