# Peer review of "Peanut Shell Extract Improves Mitochondrial Function in db/db Mice via Suppression of Oxidative Stress and Inflammation"

_nutrients, 2024, doi:10.3390/nu16131977_

Round 1

Reviewer 1 Report

Comments and Suggestions for Authors

The study of Deshmukh et al. is interesting and innovative. It provides a way to recycle peanut shells that usually end up in the bin. However, I have some concerns about the gaps in this study. Authors should make specific connection between the sections on how or why experiments were performed. These will make the history esay to follow.

It would be interesting to establish the mechanism of PSE, how it decreases stress, or increases mitochondrial function. I think an illustration in the discussion will help. Regarding mitophagy, at least some mitochondrial markers need to be studied (TOMM20, TIMM23 or…, etc). An increase in PINK1 level is not sufficient to discuss mitophagy induction. Moreover, based on the decrease of PINK1 upon treatment, one could think that PSE inhibits or reduces mitophagy. What about LC3, LAMP1/2? Have you tested PSE on cell culture in an oxidative stress model.

 I miss the study of some antioxidant enzyme levels (HO-1, Catalase, etc.). Undoubtedly, this would confirm the activity of NRF2. Is it possible to measure or detect ROS somehow in tissue samples to determine whether the NRF2 increase was able to mitigate ROS production?

This part of inflammation is fascinating, and it would be interesting for the readers to understand why the authors are investigating IBA1 in liver tissue and WAT, a marker that is mainly expressed in microglia. It lacks contextualization and arguments. What is the role of IBA1 in non-neuronal tissues?

In the discussion, it seems like there is a direct relation between NRF2 and PGC1, and the experiments done here are not strong enough to support this link.

Why authors are using in all the data an Uncorrected Fisher's LSD, was it difficult to obtain a significant variation.

It seems like the brain and WAT are more susceptible to changes than the liver. Do you have any arguments for that? Furthermore, what do you think about the opposite variation of DRP1 and FIS1? And why FIS1 result is missing in the discussion.

Thank for your comprhenesion.

Good Luck.

Comments on the Quality of English Language

The english is good. The text needs miner correction.

Author Response

Dear Reviewer,

Thank you so much for the thorough review of our manuscript.  We have carefully addressed comments made by reviewers.  Please see our response below with changes in highlighted yellow in the revised manuscript attached.

We believe that we have responded to all reviewers’ concerns and comments, and look forward to your favorable decision.

Thank you

Reviewer 2 Report

Comments and Suggestions for Authors

This paper investigated the significance of peanut shell extract (PSE) in improving mitochondrial function in db/db mice by inhibiting oxidative stress and inflammation. However, there are several points in the paper that need to be revised and improved, such as the batch numbers of the reagents used in the study should be provided; the grade and qualification number of the mice used in the study should be provided; the license number of the laboratory should be provided; the introduction should be appropriately compressed and the methodology should be simplified; and why there is no positive control group? What is the number of animals in each group, 6 for control and 14 for treatment? Why was the number of animals inconsistent? Different strains of mice as a control, comparability is a bit low; all the results of the graphs and so on should have a sample size, which is the basis of statistics; references should be based on the last three years, the format is uniform.

Author Response

Dear Reviewer,

Thank you so much for the thorough review of our manuscript. We have carefully addressed comments made by reviewers.  Please see our response below with changes in highlighted yellow in the revised manuscript attached.

We believe that we have responded to all reviewers’ concerns and comments and look forward to your favorable decision.

Thank you

Reviewer 3 Report

Comments and Suggestions for Authors

In this study authors evaluated the effects of dietary peanut shell extract (PSE) supplementation on mitochondrial function and anti-oxidative stress/inflammation in diabetic mice and found that PSE supplementation improved mitochondrial function in the brain, liver, and WAT of diabetic mice suppressing oxidative stress and inflammation.

The manuscript is interesting, generally well written and well illustrated. Only minor points deserve to be improved.

Authors affiliations: Authors must follow the journal style

Tables: reduce the characters size according to the journal style

Table 2: Product codes must be added

Figure 1, WAT: The quality of this blot should be improved 

Line 364: Correct "NRFA" with NRF2

Lines 364-367: "....NRF2 dissociates from Keap1 (a sensor for oxi- 365 dative and electrophilic stresses) into the nucleus; consequently, NRF2 loses its ability to 366 induce detoxification and antioxidant enzymes......" This is not correct since under oxidant stimuli NRF2 detachs from KEAP1 and migrates into the nucleus to bind the ARE sequences in the promoter region of antioxidant enzymes. Moreover, the multifaceted role of this pathway deserves to be highlighted since it plays a key role in cancerous and non cancerous diseases (see PMID: 37296665, PMID: 35901941).  

An accurate revision of typing errors is recommended 

Author Response

(The authors gave the same response as above.)

Round 2

Reviewer 1 Report

Comments and Suggestions for Authors

Accepted

Reviewer 2 Report

Comments and Suggestions for Authors

This paper has certain research significance, writing standard, agreed to publish.

Comments on the Quality of English Language

The language expression is accurate and standardized.